# A Novel DNA Variant in *SMARCA4* Gene Found in a Patient Affected by Early Onset Colon Cancer

**DOI:** 10.3390/ijms25052716

**Published:** 2024-02-27

**Authors:** Federica Di Maggio, Giuseppe Boccia, Marcella Nunziato, Marcello Filotico, Vincenzo Montesarchio, Maria D’Armiento, Francesco Corcione, Francesco Salvatore

**Affiliations:** 1CEINGE-Biotecnologie Avanzate Franco Salvatore, 80145 Naples, Italy; dimaggio@ceinge.unina.it (F.D.M.); nunziato@ceinge.unina.it (M.N.); 2Department of Molecular Medicine and Medical Biotechnologies, University of Naples “Federico II”, 80131 Naples, Italy; 3Department of Public Health, University of Naples “Federico II”, 80131 Naples, Italy; giuseppe.boccia@unina.it (G.B.); dottmarcellofilotico@gmail.com (M.F.); 4Division of Medical Oncology, AORN dei Colli-Monaldi Hospital, 80131 Naples, Italy; vincenzo.montesarchio@ospedalideicolli.it; 5Pathology Unit, Department of Public Health, University of Naples “Federico II”, 80131 Naples, Italy; maria.darmiento@unina.it

**Keywords:** early cases of EOCRC, colon cancer, multi-gene oncology panel, NGS, *SMARCA4* pathogenic mutation

## Abstract

Colorectal cancer is the third leading cause of death from neoplasia worldwide. Thanks to new screening programs, we are now seeing an increase in Early Onset of ColoRectal Cancer (EOCRC) in patients below the age of 50. Herein, we report a clinical case of a woman affected by EOCRC. This case illustrates the importance of genetic predisposition testing also in tumor patients. Indeed, for our patient, we used a combined approach of multiple molecular and cellular biology technologies that revealed the presence of an interesting novel variant in the *SMARCA4* gene. The latter gene is implicated in damage repair processes and related, if mutated, to the onset of various tumor types. In addition, we stabilized Patient-Derived Organoids from the tumor tissue of the same patient and the result confirmed the presence of this novel pathogenic variant that has never been found before even in early onset cancer. In conclusion, with this clinical case, we want to underscore the importance of including patients even those below the age of 50 years in appropriate screening programs which should also include genetic tests for predisposition to early onset cancers.

## 1. Introduction

Among the common malignancies, colorectal cancer (CRC) is the third leading cause of death in both men and women worldwide [1]. The incidence of CRC has increased in low- and middle-income countries throughout the past few decades, whereas its incidence has been steady in high-income countries [2]. Thanks to the implementation of screening programs that are progressively more registered in industrialized countries, it is becoming increasingly possible to detect colon cancer at early stages [3]. Furthermore, we are seeing an increase in case numbers in subjects under the age of 50 years, considered as “early onset CRC’’ (EOCRC) which accounts for 10–12% of all CRCs [4,5]. In addition, the incidence and mortality rates in individuals younger than 50 years of age are also increasing compared with those in individuals older than 50 years of age [6]. This case-mix analysis, which was chosen from a larger group of CRC patients because of their early onset, highlights the growing need to study the younger population to understand better the mechanisms possibly involved, but not yet completely known, such as the analysis at the genomic level, at an earlier age, particularly in familial cases, and also by studying genomic predisposition.

Unfortunately, to date, there are no clear factors that may be indicated for EOCRC initiation, but conventional environmental risk factors and no specific genetic factors seem to be implicated in the pathogenesis of this form [7]. Some risk factors related to lifestyle are: smoking, alcohol abuse, obesity, high consumption of red meat and a sedentary lifestyle with little physical activity may contribute to the onset of CRC in general [8,9].

Moreover, efforts are being made to understand the molecular basis of genomic predisposition of EOCRC, as about 65% of these are considered sporadic forms, about 13–33% have a family history of CRC, and 22–10% of these EOCRC patients result carriers of variants at germline level in genes not strictly related to the predisposition of colorectal cancers (i.e., *BRCA1*, *BRCA2*, *POLE*, *PTEN*, *TP53*, *ATM*) [5,7,10]. Important aspects to be considered are, in addition, the three main pathways of carcinogenesis that are involved and altered in the onset and development of CRC: microsatellite instability (MSI), chromosomal instability (CIN), and CpG islet methylating phenotype (CIMP) [8,11]. Therefore, in the last few studies, conducted to understand the differences between the early and late onset of CRC, some authors have focused on molecular aspects, using DNA sequencing, and epigenetic modification (particularly DNA methylation) [5].

Therefore, it was seen that EOCRCs tend to be microsatellite stable (MSS), with a loss of the microsatellite instability-high (MSI-H) profile compared to late-onset CRC [4]. Moreover, EOCRCs have alterations at various chromosomal levels (i.e., chromosomal rearrangements and/or gene deletion/amplification), and not always in genes considered more specific to colon cancer [12]. In addition, the genomes of these subjects appear to be euploid and have a higher level of hypomethylation than subjects with late-onset CRC [12,13]. Another aspect to consider in patients with EOCRC is the histology of the tumor tissue. In fact, they are tumors with signet ring cells, often with poor differentiation or with features of perineural or venous invasion [9,14]. Unfortunately, no real precise association has been found among all these factors and the onset of EOCRC, probably because the data now available are considerably heterogeneous within this group.

Therefore, the study of patients under the age of 50 years affected by colon cancer should be studied by multidisciplinary teams, in order to understand the molecular mechanisms underlying colon cancers at an early age. Another important aspect to consider is that these patients often have molecular features that are not “typical” of CRC [15]. Indeed, mutations at the germline level in EOCRC patients are found in genes not typically related to CRC in general and therefore, these putative patients may be included in differentiated screening to avoid missing genetic information that could aid their management.

## 2. Case Presentation and Personal History

### 2.1. Clinical Case Report

The patient was a 43-year-old female, who, at the time of diagnosis, was found to be affected by left-colon cancer. The patient in May 2021 was admitted to the hospital for abdominal colic, afterwards, the patient underwent a colonoscopy which revealed a vegetative and ulcerated easily bleeding neoformation at sigma level arranged as a sleeve around the lumen, which appears narrowed; (computed tomography) CT scan reports no sites of metastasis.

Considering the young age of the onset of CRC in the patient, she, also underwent genetic counseling and was included in a research program ongoing at “CEINGE-Advanced Biotechnology-Franco Salvatore”(research institute in Naples, Italy) to investigate possible variants in genes associated with the onset of colorectal, breast, and ovarian, and prostate cancers [16,17]. During genetic counseling, the patient reported no other lesions in other organs or tissues but reported familiarity with oncological diseases. Her father was affected by a kidney tumor, and her maternal grandfather by colon cancer (Figure 1A and Table 1). Indeed, Table 1 is the patient’s entire anamnesis in a schematic manner to aid comprehension of the patient’s prospective risk factors, as well as the diagnostic investigations and therapies carried out.

After about 10 days, the patient underwent surgery and a laparoscopic left hemicolectomy was performed. As part of a research project, informed written consent was obtained from the patient according to the procedure established by the Second Helsinki Declaration and in compliance with Italian and local regulations (Ethics Committee 318/20).

A conventional pattern for sampling was followed. Therefore, two EDTA tubes of blood were drawn the day before the surgery; whereas, at the time of the operation, small pieces of tumor and healthy adjacent tissues were taken to perform molecular and cellular research investigations (Figure 1B).

The lesion was described as poorly differentiated colonic adenocarcinoma (G3), ulcerated, with areas of necrosis and hemorrhagic, with solid aspects infiltrating the tissue wall up to the adipose tissue. Immunostaining for chromogranin and synaptophysin was also performed, which were negative; instead, the Ki67 cell proliferation index (clone 30-9, Ventana, Roche Diagnostics International, Rotkreuz, Switzerland) was about 80% (See Table 1).

Three months after the surgical procedure the patient underwent adjuvant antiblastic treatment according to the XELOX scheme [18].

### 2.2. Genetic Analysis

Given the early onset of colorectal cancer and the familiarity with oncological diseases, we studied this case in greater detail since the possible causes or predisposition of EOCR are still not clear. Therefore, we used an approach that included multiple genetic analyses to study as comprehensively as possible this case: (I) search for putative susceptible variants for colorectal cancer onset by using customized multi-gene panel [16,17,19], (II) comparison of variants found in the above-mentioned panel (gDNA) derived from blood, tumor tissue, and adjacent healthy mucosa, (III) study of microsatellite instability (MSI), (IV) study of copy number variants through multiplex ligation-dependent probe amplification (MLPA) and (V) Patient’s derived organoid (PDO) stabilization (see Figure 1C–E). In addition, samples from parents, children, and living relatives were requested but, unfortunately, they declined. For the multiple methodologies used in this paper, see the Section 4.

## 3. Results & Discussion

The sequencing run produced excellent results for each type of sample analyzed (p1_T0, p1_K, p1_H, and p1_O) (see Section 4.2): the library prepared to start from peripheral blood (p1_T0) produced 2.700.076 reads, 94.7% of which passed the quality filters. The average coverage obtained is 408× in the target regions and 97.5% of the entire sample is covered by at least 20×; p1_K produced 1.137.043 reads, and 98.2% passed all the quality filters; the average read depth was 104× and 99.06% of target regions reached at least 20×; p1_H obtained 1.130.438 reads of which 98% passed quality filters; the average read depth was 98× and 99% of target regions were covered by at least 20×; finally, p1_O obtained 1.164.664 reads, of which 98.2% passed quality filters; the average read depth was 115× and 99.15% of target regions were covered by at least 20×.

In all the samples (p1_T0, p1_K, p1_H, and p1_O) analyzed with a multi-gene panel, we identified a variant in the *SMARCA4* gene. The variant is exonic and is nonsense that involves the formation of a premature stop codon not reported in clinical databases as ClinVar and dbSNP: c.3854T>A, p.Leu1285Ter on the transcript NM_001128849.2 (36 exons). The variant produces a premature stop codon at exon 27, causing the synthesis of a truncated protein (Figure 2). Direct sequencing was also used to confirm the variant (Sanger sequencing). We then performed a bioinformatic prediction analysis with the Franklin database (https://franklin.genoox.com, Franklin by Genoox), which reports this variant as likely pathogenic according to the American College of Medical Genetics and Genomics (ACMG) classification; instead, the CADD score is reported to be 41 for the above chromosomal position.

Another aspect that we assessed in this clinical case was microsatellite instability. We performed this molecular investigation on gDNA extracted from blood, tumor tissue, and PDOs of the patient, and in all three cases, it was considered a microsatellite-stable (MSS) tumor as there were no loci with instability. This finding is also in line with what has been reported in the literature as far as microsatellite instability in EOCRC is correlated with low instability [9,20]. Microsatellites (MS) are short tandem repeated DNA sequences (1–6 bases), distributed across the human genome, that change in length when there is a mismatch repair defect [21,22]. 

Therefore, MSI is due to inefficient DNA mismatch repair (MMR), which then leads to the accumulation of errors during the DNA replication phase [22,23]. Indeed, considering the recent updates for the use of immunotherapy drugs in patients with advanced colorectal cancer (dMMR/MSI-H), it is becoming increasingly important to know the MS state of the tumor for better patient management and, thus, for the use of increasingly targeted and personalized drug therapies.

For a better comprehension of the case, we further studied six different genes, which are among the most common load-bearing CNVs, (see Section 4.4) through MLPA, and we found there were no copy number alterations in the above genes. This approach was chosen because some correlation of CNVs with CRC progression is also known in the literature [24].

Finally, the last investigation performed on this patient was the stabilization of the PDO (p1_O) by tumor cells taken at the time of surgery. Stabilization of this 3D in vitro model first gave us the opportunity to assess the growth of the tumor cells in culture, and then to compare the genome of PDOs with that of the tumor tissue in order to assess their correspondence. This was done through sequencing analyses of the 56-multi gene panel, by which we also found in p1_O the variant in the *SMARCA4* gene. In addition, using the multi-gene panel, we found a somatic variant in p1_O in the *PIK3CA* gene. The variant found is annotated in the ClinVar database, c.2176G>A, p.Glu726Lys on the transcript NM_006218.4, as pathogenic. The variant falls in exon 14, and the single nucleotide change causes a missense variant, previously described as a mutational hotspot [25]. Pathogenic variants in this gene are found in approximately 20–25% of CRC [26]. In the p1_T0 sample, this mutation was not present, thus indicating that it is somatic and appears during the progression of the tumor growth. Since we were studying primarily the predisposition at the genomic germline level, we did not further discuss this variant, which was a later event. Nonetheless, the significance of this mutation is important, particularly for potential therapeutic findings, also known in the literature [26,27] that, thanks to the established PDOs, we will investigate later. We also performed an MSI evaluation that confirmed the stability (MSS) of the gDNA derived from the PDOs. Thus, with this strategy, we verified a good concordance between the tumor tissue and the stabilized PDO. This certainly opens up new lines of research for the use of these 3D models for increasingly personalized and precision medicine [28,29]. With this procedure, in the near future, we hope to be able to use a variety of drugs to test them first in vitro, on the stabilized model, and then to guide the treatment of patients based on the response obtained in organoids.

One of the most interesting results of this clinical case is the discovery of an unreported variant in the *SMARCA4* gene in early colon cancer patient; this gene is located on chromosome 19p13 and encodes for a subunit of the SWI/SNF complex [30]. The latter is implicated not only in chromatin remodeling but also in DNA repair because it is recruited to DNA damage response (DDR) sites by PARP1. Thus, it is scientifically proven to be a tumor suppressor of cancer [30,31,32]. SWI/SNF complexes consist of approximately 10–15 different protein complexes that are classified into three groups: canonical BRGI/BRM-associated factor (cBAF), polybrome-associated BAF (PBAF), and non-canonical, GLTSCR1/L-associated BAF that has been recently characterized (ncBAF or GBAF) [33]. Each complex contains an ATPase site that comprises catalytic subunits BRG1/BRM (encoded by *SMARCA4* and *SMARCA2*, respectively) and a common core [32]. To date, the function of some domains is known, but for others, further studies are still needed. Because of the above, is clear that these complexes, when mutated, are implicated in the processes of carcinogenesis, and it is well known that loss of function of one of the SWI/SNF subunits impairs DNA damage repair, normal cell cycle regulation, and defect of some signaling pathways such as PI3K/AKT/mTOR and MYC regulation. About 20% of cancers appear to be mutated in a gene that codes for this complex [32].

Therefore, germline pathogenic variants in *SMARCA4* have been reported to be related to the predisposition of several types of cancers, namely Small Cell Carcinoma of the Ovary, Hypercalcemic Type (SCCOHT), early onset Ovarian Cancer (OC), neuroblastoma, and rhabdoid tumor predisposition syndrome [34,35,36]. Germline mutations at *SMARCA4* that cause a truncated protein are reported to be the key genetic event because a decrease in functional BRG1 protein is expected and has often been related to such cancer pathologies as SMARCA4-deficient Non-Small-Cell-Lung Cancer (NSCLC) and rhabdoid tumor predisposition syndrome [35,37,38,39]. Indeed, pathogenic variants in the *SMARCA4* gene are mainly associated with early onset (≤40 years) OC [36]. Moreover, studies have been conducted to understand possible gender differences, and indeed, it appears that Overall Survival (OS) between male and female genders in patients with mut*SMARCA4* carcinoma is worse in male patients than in female patients [40]. These data could be useful for prognostic stratification in clinical practice.

Even though our investigation is focused on a single case, the germline nonsense mutation found in *SMARCA4* is of significant interest because it is the first time it has been described. Furthermore, germline mutations in the *SMARCA4* gene result in Rhabdoid Tumor Predisposition Syndrome type 2 (RTPS2), which predisposes to different types of cancers (i.e., brain, spine, lung, bladder, pelvis, kidney, and ovaries) [35,36,41,42]. Recognizing germline *SMARCA4* mutations could thus benefit family testing and risk mitigation, especially since their involvement in carcinogenesis is still unknown.

In contrast to what has been examined so far in the association between pathogenic mutations in *SMARCA4* and early onset in SCCOHT and NSCLC, there are currently no close correlations in the literature with this gene and CRC, particularly in young people (EOCRC), therefore necessitating further investigation to find potential linkages [37]. However, variants in genes not closely related to colon cancer are instead to be considered with more interest in patients with early onset, as different studies are being conducted to better understand the molecular basis of juvenile forms of CRC [5], which is important knowledge for therapeutic assets.

Finally, with this case, we hope to contribute to a better understanding of cases of EOCRC. Indeed, more research is required to determine the molecular bases that regulate the early onset of colon cancer. Notably, patients below 50 years of age are not included in screening programs for colon cancer, as the contrary occurs in the case of more well-known cancers such as breast and ovarian [43]. Moreover, for these patients and for other cancers where early presentation may appear, there is currently no established screening program for searching germline variants in susceptibility genes, thus indicating the need to use, at least genes known since now, that could help in patient management and treatment. From this perspective, the patient studied seems to us, in this context, to be of interest, as truncated variants in a gene correlated with tumor occurrence, but not strictly associated with the most studied in CRC have never been reported. Therefore, in cases with earlier forms, as in our patient, a panel with few genes in early onset cases of cancer may be relevant for a more precocious diagnostic and then therapeutic asset.

In conclusion, this case report could represent a good starting point for a larger knowledge of altered genes in a greater cohort of early onset subjects to better understand the possible effects and incidence of this novel variant.

## 4. Materials and Methods

### 4.1. Organoid Stabilization

We took a piece of tumor tissue of about 1 cm^2^ from the enrolled patient to proceed with organoid stabilization. Therefore, once the tumor tissue reached the laboratory, transported to us at 4 °C inside the medium Advanced+++ (see Appendix A), we immediately started the digestion phase. First, we performed a wash of the tumoral tissue using a broad-spectrum of antibiotics with Penicillin-Streptomycin (Thermo Fisher Scientific, Waltham, MA, USA) at 10,000 U/mL and Primocin (Invivogen, San Diego, CA, USA) at 50 mg/mL for a total of 5 min at room temperature. Then, we carried out the digestion, the split, and the organoid culture condition protocols as also previously described [44,45]. At the end of each split step, a check was made by microscope to assess the growth of PDOs. All the images of live PDOs’ cultures were obtained with a Leica DMI4000b (Wetzlar, Germany) inverted microscope (See Figure 1C–E). Next, we chopped the tissue into small fragments and performed two different digestion steps. After carrying out several washing and centrifugation steps, the cell pellet was resuspended in a varying amount of matrigel (Corning, New York, NY, USA), about 500 μL, and it was plated in a 24- or 48-multiwell (Sarstedt, Numbrecht, Germany). Finally, we coated the cells with the Culture Medium (see Appendix A) and we cultured them in a cell culture incubator at 37 °C and 5% CO_2_. The Culture Medium was changed every 3 days, and every 7 days was performed a split-step. During the split step, we detached the matrigel domes, and subsequently, we separated the cells with TrpleE. Lastly, we carried out the washings and centrifugation steps, after which, the pellet was resuspended in an appropriate amount of matrigel and re-plated the cells as also described above [45].

### 4.2. DNA Extraction and Sequencing Analysis

Once the blood reached the laboratory, gDNA extraction from 300 µL of peripheral blood (p1_T0) using the Promega 16 LEV blood DNA purification kit and Maxwell instrument was performed (Promega Corporation, 2800 Woods Hollow Road, Madison, WI, USA), according to the manufacturer’s instructions. Instead, in the case of fresh frozen tissues taken during surgery, gDNA extraction starting from 10 mg of each tissue was also performed. The tissues were first homogenized by using the TissueLyser (Qiagen, Hilden, Germany), as indicated in the manual, and then the resulting solution was used for gDNA extraction from tumor tissue (p1_K) and adjacent healthy mucosa (p1_H) using the Promega 16 LEV blood DNA purification kit and Maxwell instrument according to the manufacturer’s instructions. Then, each gDNA was evaluated for quantity through QuBit 4.0 fluorimeter (Thermo Fisher Scientific, MA, USA) while for the quality we used the Genomic DNA assay ScreenTape System using 4200 TapeStation (Agilent Technologies, Santa Clara, CA, USA). Instead, gDNA from patient-derived organoids (PDOs, p1_O) was extracted from 5 matrigel domes using the DNeasy Blood & Tissue Kits (Qiagen, Hilden, Germany) according to the manufacturer’s instruction.

For each gDNA (p1_T0, p1_K, p1_H, p1_O), we prepared individual libraries by enriching the gDNA with specific probes, as the panel was customized and designed in our laboratory, and the SureSelect XT Target Enrichment System (Agilent Technologies, Santa Clara, CA, USA) was used following the manufacturer instructions. Once the libraries were prepared, qualitative and quantitative assessments were carried out using the D1000 ScreenTape System using 4200 TapeStation to verify the presence of the right DNA profile and the QuBit 4.0, respectively. The libraries were thus prepared, diluted at 10 nM and then pooled together and brought to a concentration of 4 nM in a single tube. Specifically, 1 paired-end (PE 150 × 2) sequencing run was carried out on the Miseq platform (Illumina, CA, USA) using the MiSeq^®^ Reagent Kit v2 standard (300 cycles). Ten pM of the final pool were combined to 10% of 10 pM PhiX and were loaded into the MiSeq reagent cartridge, according to the manufacturer’s instructions.

The data obtained were then analyzed using Alissa software v.5.4.2) (Agilent Technologies, Santa Clara, CA, USA) [17]. We used the Alissa Align and Call tool to align short reads with the reference sequence of the Genome Reference Consortium Human Build 37 (GRC-hg37), determine quality control metrics, and detect genetic variations. Then, we used Alissa Interpret to annotate the variant and to prioritize putatively causative variants. We filtered the variants based on the above parameters: quality, allele frequency < 0.01 based on ExAC database, genomic localization (exonic, intronic, UTR3′, UTR5′), and finally based on possible implications of the variants with disease occurrence considering the clinical significance (benign, likely benign, uncertain significance, likely pathogenic and pathogenic variants) using the ClinVar database [46]. Variants not reported in ClinVar were further annotated using the Franklin database, according to the ACMG classification [47], and CADD scores were evaluated [48] to assess any potential additional pathogenicity. Finally, the variants found to be potentially pathogenic were then confirmed by Sanger sequencing [49]. Specific primers to enrich the region of interest were designed using Primer3 [50] and Primer Blast [51]. We then amplified the region of interest and by Sanger sequencing confirmed the presence of the variant.

### 4.3. Microsatellite Instability Evaluation

Given the current prognostic and predictive role of MSI in colorectal cancer, we performed this investigation to better understand the genetic asset of our patient. Therefore, MSI status was studied by PCR amplification of DNA sequences; the instability was determined by comparing the length of nucleotide repeats in tumor cells and in a paired normal sample obtained from the same patient [52]. We analyzed 10 markers (*BAT25*, *BAT26*, *D2S123*, *D17S250*, *D5S346*, *BAT40*, *D18S58*, *NR21*, *NR24*, and *TGFβRII*) using the Titano MSI kit (Diatech Pharmacogenetics S.R.L., Jesi, Italy). We diluted the gDNA samples isolated from blood (p1_T0), tumoral tissue (p1_k1), and organoid (p1_O1) at 5 ng/μL and used a total of 75 ng of each DNA for the amplification of each sample through multiplex-PCR with fluorophore-labeled primers to obtain fragments including all markers of interest, according to the manufacturer’s instructions. Then, for fragment analysis, we ran the fragments obtained in the previous step, for each sample, on an automated sequencer (ABI3130-XL Genetic Analyzers; Applied Biosystems, Waltham, MA, USA), following the manufacturer’s instructions. We then compared each profile obtained for each sample analyzed with the other (p1_T0 vs. p1_k1 vs. p1_O1) using Genemapper software (v.6) following the manufacturer’s instructions.

### 4.4. Multiplex Ligation-Dependent Probe Amplification (MLPA) for Gene Copy Number Analysis

To determine whether there were any alterations related to copy number, we performed analyses of multiplex ligation-dependent probe amplification (MLPA) to test 6 genes (*BRCA1*, *BRCA2*, *PALB2*, *RAD50*, *RAD51C*, *RAD51D*) using 3 different SALSA MLPA Probemix: P002, P090, P260 (MRC Holland, Amsterdam, The Netherlands). Starting from 75 total ng of gDNA (p1_T0), we performed the entire procedure following the manufacturer’s instructions. Fragment separation was performed by an automated sequencer (ABI3130-XL Genetic Analyzers). The generated raw data were then analyzed with Coffalyser software (v.220513.1739) according to the manufacturer’s instructions.

## Figures and Tables

**Figure 1 ijms-25-02716-f001:**
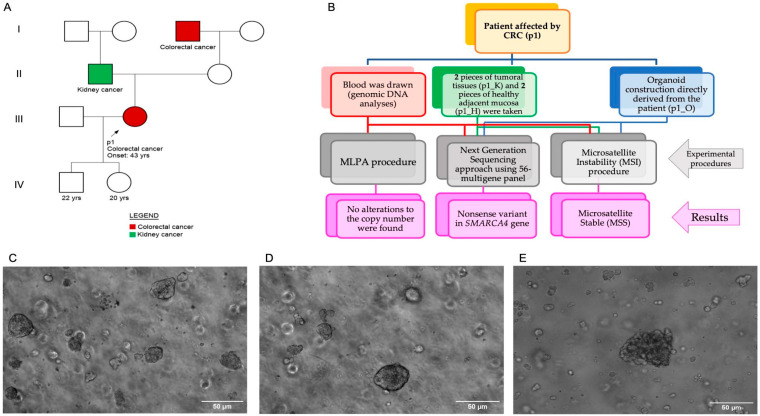
(**A**) Pedigree of the entire family of the proband. The proband is a 43-year-old woman considered an early onset bearing colorectal cancer. The patient has familiarity with oncological pathologies as her father had kidney cancer, while her maternal grandfather had colorectal cancer. (**B**) For the enrolled patient, a specific scheme was followed for the collection of biological samples. During genetic counseling, 2 EDTA-blood samples were taken (red box). Furthermore, pieces of tumor tissue and adjacent healthy mucosa were taken at the time of surgery (green box). Again, at the time of surgery, a full-thickness piece of tumor tissue was also taken for subsequent stabilization of the organoid (blue box). From each sample, we extracted gDNA and carried out different molecular biology procedures, as described in the Material and Method section. (**C**–**E**) Images of patient-derived organoids (PDOs) stabilized from the patient’s tumor tissue. Before carrying out the split phase, we always collected the images of the organoids; therefore, the figures show the images consecutively taken from passage 1 (**C**) to passage 3 (**E**). The images were taken with a Leica DMI4000b inverted microscope.

**Figure 2 ijms-25-02716-f002:**
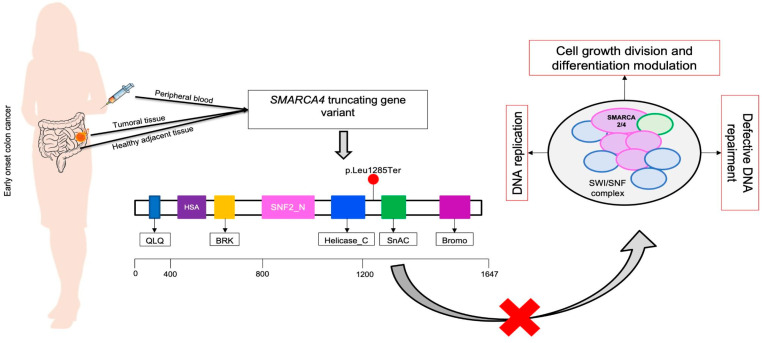
Graphical representation and functions of the *SMARCA4* gene. The variant in the *SMARCA4* gene was found in DNA extracted from peripheral blood (p1_T0), tumor tissue (p1_K), healthy tissue (p1_H) and organoid (p1_O). The variant leads to the formation of a truncated protein whose multiple biological functions are defective in certain pathways specific to the SWI/SNF complex in which *SMARCA4* is implicated. So, the protein complex’s proper function is inhibited, as indicated by the red cross symbol.

**Table 1 ijms-25-02716-t001:** Anamnestic information and the diagnostic procedure carried out for the proband.

General Information									
Patient ID	Sex	Age at the diagnosis	Obesity	Smoke	Physics activities	Food intolerances	Age of menarche	Pregnancies	Familiarity with oncological diseases
p1	Female	43 y	NO	NO	NO	NO	11 y	Yes (*n* = 2)	Father (K kidney); maternal grandfather (K colon cancer)
**Diagnostic and therapeutic procedures on the proband**									
Patient ID	Symptoms	Abdominal CT scan with contrast medium	Colonoscopy	Surgery	Histological report		Immunoblotting	Pathological stage (TNM ^#^ and grading)	Genetic analyses
p1	Abdominal colic with diarrhea	Left colon neoplasia with no other sites of neoplasia	Ulcerated and bleeding at sigma level lesion	Left hemicolectomy through laparoscopic approach	Poorly differentiated, ulcerated adenocarcinoma of the colon with areas of necrosis and hemorrhagic, with solid aspects infiltrating the wall up to the adipose tissue. Low-grade tumor budding, occasional images of endovascular neoplastic embolization, absence of perineural invasion.		Chromogranin and synaptophysin were negative	T3N0Mx; G3	Analysis of multi-gene (*n* = 56) panel correlated with the onset of oncological diseases, and evaluation of microsatellite instability

^#^ Tumor staging is carried out following the TNM (Tumor, Node, Metastasis) system. Each letter will be followed by a number that will define: (T) the size of the primary tumor, (N) the involvement of the regional lymph nodes, and (M) the presence of distant metastases.

## Data Availability

The sequencing data will be available upon request by users to the corresponding author.

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
