# Peer review of "A Novel DNA Variant in SMARCA4 Gene Found in a Patient Affected by Early Onset Colon Cancer"

_ijms, 2024, doi:10.3390/ijms25052716_

Round 1
Reviewer 1 Report
Comments and Suggestions for Authors
Here the authors report a clinical case of a 43-year-old female patient with early-onset colorectal cancer (EOCRC), who was found to have a novel nonsense variant in the SMARCA4 gene, which encodes a subunit of the SWI/SNF chromatin remodeling complex involved in DNA repair and tumor suppression. The authors used a combination of molecular and cellular techniques, including a customized multi-gene panel, microsatellite instability analysis, multiplex ligation-dependent probe amplification, and patient-derived organoid stabilization, to characterize the genetic and phenotypic features of the patient’s tumor and compare them with normal tissue and blood samples. The article suggests that the SMARCA4 variant may be a predisposing factor for EOCRC, as it has been associated with other early-onset cancers, such as ovarian and lung cancers. They also highlight the importance of including patients younger than 50 years in screening programs and genetic tests for early detection and personalized treatment of EOCRC.
Some possible good and bad points of this article are:
Main points:
The article presents a rare and interesting case of EOCRC with a novel pathogenic variant in a gene not typically related to CRC.
The article uses a comprehensive and multidisciplinary approach to investigate the molecular basis and the biological behavior of the patient’s tumor.
The article contributes to the growing knowledge of the genetic and epigenetic factors involved in EOCRC and provides insights for future research and clinical practice.
Minor points:
The article is based on a single case report, which limits the generalizability and the statistical significance of the findings. Could the author discuss this further, by for example, talking about SMARCA4 mutations in other diseases.
The article does not discuss the possible interactions or synergies between the SMARCA4 variant and other genetic or environmental risk factors for EOCRC. Could this be added to the discussion?
Comments on the Quality of English Language
Minor typos. Please double check the text.
Author Response
A novel DNA variant in SMARCA4 gene found in a patient affected by early onset colon cancer
Federica Di Maggio1,2*, Giuseppe Boccia3*, Marcella Nunziato1,2, Marcello Filotico3, Vincenzo Montesarchio4, Maria D’Armiento5, Francesco Corcione3#, Francesco Salvatore1,2#
1 “CEINGE-Biotecnologie Avanzate Franco Salvatore”, Naples, Italy; dimaggio@ceinge.unina.it, nunziato@ceinge.unina.it, salvator@unina.it
2 Department of Molecular Medicine and Medical Biotechnologies, University of Naples “Federico II”, Naples, Italy; dimaggio@ceinge.unina.it, nunziato@ceinge.unina.it, salvator@unina.it
3 Department of Public Health, University of Naples “Federico II”, Naples, Italy; giuseppe.boccia@unina.it; dottmarcellofilotico@gmail.com; francesco.corcione@unina.it
4 Division of Medical Oncology, AORN dei Colli-Monaldi Hospital, 80131 Naples, Italy. vincenzo.montesarchio@ospedalideicolli.it
5 Department of Public Health, Pathology Unit, University of Naples "Federico II", Naples, Italy. maria.darmiento@unina.it
*Co-first authors contributed equally to the work
#Co-last and co-corresponding authors
Reviewer 1:
Comments:
Here the authors report a clinical case of a 43-year-old female patient with early-onset colorectal cancer (EOCRC), who was found to have a novel nonsense variant in the SMARCA4 gene, which encodes a subunit of the SWI/SNF chromatin remodeling complex involved in DNA repair and tumor suppression. The authors used a combination of molecular and cellular techniques, including a customized multi-gene panel, microsatellite instability analysis, multiplex ligation-dependent probe amplification, and patient-derived organoid stabilization, to characterize the genetic and phenotypic features of the patient’s tumor and compare them with normal tissue and blood samples. The article suggests that the SMARCA4 variant may be a predisposing factor for EOCRC, as it has been associated with other early-onset cancers, such as ovarian and lung cancers. They also highlight the importance of including patients younger than 50 years in screening programs and genetic tests for early detection and personalized treatment of EOCRC.
Some possible good and bad points of this article are:
Main points:
The article presents a rare and interesting case of EOCRC with a novel pathogenic variant in a gene not typically related to CRC.
The article uses a comprehensive and multidisciplinary approach to investigate the molecular basis and the biological behavior of the patient’s tumor.
The article contributes to the growing knowledge of the genetic and epigenetic factors involved in EOCRC and provides insights for future research and clinical practice.
Minor points:
- The article is based on a single case report, which limits the generalizability and the statistical significance of the findings. Could the author discuss this further, by for example, talking about SMARCA4 mutations in other diseases.
Reply: Thank you, we modify the text accordingly.
See in text at page7-8; lines 257-263:” Even though our investigation is focused on a single case, the germline nonsense mutation found in SMARCA4 is of significant interest because it is the first time it has been described. Furthermore, germline mutations in the SMARCA4 gene result in Rhabdoid Tumor Predisposition Syndrome type 2 (RTPS2), which predisposes to different types of cancers (i.e. brain, spine, lung, bladder, pelvis, kidney, and ovaries) [35,36,41,42]. Recognising germline SMARCA4 mutations could thus benefit in family testing and risk mitigation, especially since their involvement in carcinogenesis is still unknown.”
- The article does not discuss the possible interactions or synergies between the SMARCA4 variant and other genetic or environmental risk factors for EOCRC. Could this be added to the discussion?
Reply: Thank you, we now make clearer the links existing between SMARCA4 gene variant in EOCRC neoplasia not yet known.
See in the text at page 8 lines 264-267 “In contrast to what has been examined so far in the association between pathogenic mutations in SMARCA4 and early onset in SCCOHT and NSCLC, there are currently no close correlations in the literature with this gene and CRC, particularly in young people (EOCRC), therefore necessitating further investigation to find potential linkages [37].”

Reviewer 2 Report
Comments and Suggestions for Authors
Comments to the manuscript ijms-2892934. This is an interesting, well-written and structured document in which for the first time the presence of this novel pathogenic variant that has never been found before even in early-onset cancer is described. Only small comments are suggested below.
1. Table 1 can be described in the text
2. standardize the writing of references in accordance with the journal's guidelines
Author Response
A novel DNA variant in SMARCA4 gene found in a patient affected by early onset colon cancer
Federica Di Maggio1,2*, Giuseppe Boccia3*, Marcella Nunziato1,2, Marcello Filotico3, Vincenzo Montesarchio4, Maria D’Armiento5, Francesco Corcione3#, Francesco Salvatore1,2#
1 “CEINGE-Biotecnologie Avanzate Franco Salvatore”, Naples, Italy; dimaggio@ceinge.unina.it, nunziato@ceinge.unina.it, salvator@unina.it
2 Department of Molecular Medicine and Medical Biotechnologies, University of Naples “Federico II”, Naples, Italy; dimaggio@ceinge.unina.it, nunziato@ceinge.unina.it, salvator@unina.it
3 Department of Public Health, University of Naples “Federico II”, Naples, Italy; giuseppe.boccia@unina.it; dottmarcellofilotico@gmail.com; francesco.corcione@unina.it
4 Division of Medical Oncology, AORN dei Colli-Monaldi Hospital, 80131 Naples, Italy. vincenzo.montesarchio@ospedalideicolli.it
5 Department of Public Health, Pathology Unit, University of Naples "Federico II", Naples, Italy. maria.darmiento@unina.it
*Co-first authors contributed equally to the work
#Co-last and co-corresponding authors
Reviewer 2:
Comments:
This is an interesting, well-written and structured document in which for the first time the presence of this novel pathogenic variant that has never been found before even in early-onset cancer is described. Only small comments are suggested below.
- Table 1 can be described in the text
Reply: Thank you for this suggestion. We would prefer to retain the table. Indeed, it gives, at a glance, the status and anamnestic history and the procedures performed (diagnostic and therapeutic). We now make some more detailed information according to the referee suggestions.
See in the text at page 3, lines 101-104:” Indeed, in Table 1 is patient's entire anamnesis in a schematic manner to aid comprehension of the patient's prospective risk factors, as well as the diagnostic investigations and therapies carried out”
- standardize the writing of references in accordance with the journal's guidelines:
Reply: Thank you, the references are now in accordance with the guidelines

Reviewer 3 Report
Comments and Suggestions for Authors
In this article, Di Maggio et al. presented a case of a 43 years female with colorectal cancer showing a novel germline variant in SMARCA4 gene. The case is interesting and deserves to be published, however I have few comments and suggestions:
- did the authors screen for the identified SMARCA4 variant in the parents of the patients? Father? Grand father? Or other family members at risk? It would be interesting to undertake such analysis for the other family members, in order to confirm the origin of the variant: inherited or de novo.
- Why did the authors choose to perform CNV analysis by MLPA for six genes only? Wouldn’t it be more interesting to perform microarray analysis?
- Identification of a somatic PIK3CA pathogenic somatic variant. The authors decided to disregard this variant. However; I guess that further investigations and more attention are required towards this issue. Especially that PIK3CA targeted therapy is available, even if it is not FDA approved for CRC yet.
- I would like to share this paper with authors, about SMARCA4 and SMARCA2-related therapies.it could be useful for the discussion. PMID: 33144586.
- Line 46-47: Which case-mix? The idea seems unclear to me.
- Line 58: 22 to 10% ? or 2 to 10?
- English editing and correction are needed.
Comments on the Quality of English Language
English editing is required.
Author Response
A novel DNA variant in SMARCA4 gene found in a patient affected by early onset colon cancer
Federica Di Maggio1,2*, Giuseppe Boccia3*, Marcella Nunziato1,2, Marcello Filotico3, Vincenzo Montesarchio4, Maria D’Armiento5, Francesco Corcione3#, Francesco Salvatore1,2#
1 “CEINGE-Biotecnologie Avanzate Franco Salvatore”, Naples, Italy; dimaggio@ceinge.unina.it, nunziato@ceinge.unina.it, salvator@unina.it
2 Department of Molecular Medicine and Medical Biotechnologies, University of Naples “Federico II”, Naples, Italy; dimaggio@ceinge.unina.it, nunziato@ceinge.unina.it, salvator@unina.it
3 Department of Public Health, University of Naples “Federico II”, Naples, Italy; giuseppe.boccia@unina.it; dottmarcellofilotico@gmail.com; francesco.corcione@unina.it
4 Division of Medical Oncology, AORN dei Colli-Monaldi Hospital, 80131 Naples, Italy. vincenzo.montesarchio@ospedalideicolli.it
5 Department of Public Health, Pathology Unit, University of Naples "Federico II", Naples, Italy. maria.darmiento@unina.it
*Co-first authors contributed equally to the work
#Co-last and co-corresponding authors
Reviewer 3:
Comment:
In this article, Di Maggio et al. presented a case of a 43 years female with colorectal cancer showing a novel germline variant in SMARCA4 gene. The case is interesting and deserves to be published, however I have few comments and suggestions:
- Did the authors screen for the identified SMARCA4 variant in the parents of the patients? Father? Grand father? Or other family members at risk? It would be interesting to undertake such analysis for the other family members, in order to confirm the origin of the variant: inherited or de novo.
Reply: Thank you for your suggestions. We asked whether the proband could provide some samples of the parents and her children but unfortunately, she declined.
We modify in the text accordingly the referee suggestion. See in the text at page 4, lines 148-149:” In addition, samples from parents, children, and living relatives were requested but unfortunately, they declined.”
- Why did the authors choose to perform CNV analysis by MLPA for six genes only? Wouldn’t it be more interesting to perform microarray analysis?
Reply: Thank you for your question. The MLPA was done only on the genes which are among the most common ones bearing related to copy number variants. We also thought about the array but unfortunately the machine has not been available at the moment.
See also in the text at page 6, lines 199-200:”…Which are among the most common bearing CNVs…”
- Identification of a somatic PIK3CA pathogenic somatic variant. The authors decided to disregard this variant. However; I guess that further investigations and more attention are required towards this issue. Especially that PIK3CA targeted therapy is available, even if it is not FDA approved for CRC yet.
Reply: Thank you so much, for your comment. At the time, we did not focus on this variant since we intended to highlight the germinal variants observed in SMARCA4 gene. Nonetheless, the somatic mutation in PIK3CA is undoubtedly of importance; especially because it was detected in the gDNA from the Patient's derived organoid (PDOs) and will allow us to test the chemotherapeutic effect of various drugs in PDOs. This was felt very interesting, as the referee also suggested because of their therapeutic effects.
See in the text at page 7, lines 218-220: “Nonetheless, the significance of this mutation is important, particularly for potential therapeutic findings, also known in the literature [26,27], that, thanks to the established PDOs, we will investigate later.”
- I would like to share this paper with authors, about SMARCA4 and SMARCA2-related therapies.it could be useful for the discussion. PMID: 33144586.
Reply: Thank you for the useful suggestion of the referee: we have added the reference to the paper accordingly.
- Line 46-47: Which case-mix? The idea seems unclear to me.
Reply: We selected this case and studied it from a series of colorectal cancer tissue and blood samples in a larger project. This case was chosen for its peculiarity of being an EOCRC in which the predisposition mutation was never found.
Therefore, we add the text in red; see in the text at page 2, lines 46-47:“….which was chosen from a larger group of CRC patients because of their early onset…”
- Line 58: 22 to 10% ? or 2 to 10?
Reply: Thank you for your question. The right % is 22 to 10% (we have added reference). So far, the literature reveals a range from 10 to 22% of patients with early colon cancer, who have a family history of colon cancer and may have germline variants in susceptibility genes related to cancer predisposition.
See reference number [5,7,10].
- English editing and correction are needed.
Reply: Thanks for your recommendations. An experienced author's editor has revised the text for grammar and readability.
